# Protocol for the economic evaluation of metacognitive therapy for cardiac rehabilitation participants with symptoms of anxiety and/or depression

Gemma E Shields ,[1] Adrian Wells ,[2,3] Patrick Doherty ,[4] David Reeves ,[5] Lora Capobianco ,[3] Anthony Heagerty ,[6] Deborah Buck,[1] Linda M Davies [1]

For numbered affiliations see end of article.

**Correspondence to**
Gemma E Shields;
gemma.shields@manchester.ac.uk

## ABSTRACT

**Introduction** Cardiac rehabilitation (CR) is offered to reduce the risk of further cardiac events and to improve patients' health and quality of life following a cardiac event. Psychological care is a common component of CR as symptoms of depression and/or anxiety are more prevalent in this population, however evidence for the cost-effectiveness of current interventions is limited. Metacognitive therapy (MCT), is a recent treatment development that is effective in treating anxiety and depression in mental health settings and is being evaluated in CR patients. This protocol describes the planned approach to the economic evaluation of MCT for CR patients.

**Methods and analysis** The economic evaluation work will consist of a within-trial analysis and an economic model. The PATHWAY Group MCT study has been prospectively designed to collect comprehensive self-reported resource use and health outcome data, including the EQ-5D, within a randomised controlled trial study design (UK Clinical Trials Gateway). A within-trial economic evaluation and economic model will compare the cost-effectiveness of MCT plus usual care (UC) to UC, from a health and social care perspective in the UK. The within-trial analysis will use intention-to-treat and estimate total costs and quality-adjusted life-years (QALYs) for the trial follow-up. Single imputation will be used to impute missing baseline variables. Multiple imputation will be used to impute values missing at follow-up. Items of resource use will be multiplied by published national healthcare costs. Regression analysis will be used to estimate net costs and net QALYs and these estimates will be bootstrapped to generate 10 000 net pairs of costs and QALYs to inform the probability of cost-effectiveness. A decision analytical economic model will be developed to synthesise trial data with the published literature over a longer time frame. Sensitivity analysis will explore uncertainty. Guidance of the methods for economic models will be followed and dissemination will adhere to reporting guidelines.

**Ethics and dissemination** The economic evaluation includes a within-trial analysis. The trial which included the collection of this data was reviewed and approved by Ethics. Ethics approval was obtained by the Preston Research Ethics Committee (project ID 156862). The modelling analysis is not applicable for Ethics as it will

## Strengths and limitations of this study

► A prospectively designed within-trial economic evaluation alongside a randomised controlled trial, with comprehensive data collection.
► Supplemented by an exploratory decision analytical model to assess cost-effectiveness over a longer time horizon.
► Comprehensive sensitivity analysis will be used to explore a range of alternative measures of health benefits, impact of assumptions and time horizons.
► Data limitations, such as the trial time frame, are likely to affect the plausibility of long-term results.
► Issues with generalisability due to variations across cardiac services and populations entering cardiac rehabilitation.

use data from the trial (secondary analysis) and the published literature. Results of the main trial and economic evaluation will be published in the peer-reviewed National Institute for Health Research (NIHR) journals library (Programme Grants for Applied Research), submitted to a peer-reviewed journal and presented at appropriate conferences.

**Trial registration number** ISRCTN74643496; Pre-results.

## INTRODUCTION

In the UK approximately 90 000 people start cardiac rehabilitation (CR) annually, receiving a supervised programme of care that aims to improve patients' health and quality of life.[1] The population offered CR is variable; the British Association for Cardiovascular Prevention and Rehabilitation propose three groups of priority, including acute coronary syndrome, coronary revascularisation and/or heart failure, with further groups who should be offered CR if possible (including stable angina among others).[2] The greatest number of attendees of CR in the UK come from populations with myocardial infarction, percutaneous coronary intervention, coronary artery bypass graft and heart failure.[1]

The reported benefits of CR include reduced mortality, reduced hospital admissions, improvements in patient cardiovascular risk profiles and improved psychological well-being and quality of life.[3] Around 50% of those offered CR attend.[1]

The latest figures suggest that in the UK around 18% and 28% of patients initiating CR have borderline or clinical depression and anxiety, respectively.[4] Rehabilitation programmes vary, most frequently including lifestyle risk factor management (eg, physical exercises) and health behaviour change and education.[3] Guidelines also recommend that a psychological component is included in CR care.[3] Despite this, current interventions (including pharmacological and psychosocial interventions) have been shown to have limited effectiveness in this group.[5] Psychological components of CR have been less frequently analysed in cost-effectiveness evaluations and results of cost-effectiveness studies are varied.[6] Research to determine whether psychological therapy is effective and cost-effective in CR is a priority, given recent calls for closer integration of psychological and physical health outcomes.[7]

Metacognitive therapy (MCT) may be well-suited to the needs of CR patients; it is based on the metacognitive model, which proposes that a thinking style dominated by rumination, worry and threat monitoring maintains emotional distress.[8–10] MCT is highly effective at reducing this thinking style and alleviating symptoms of depression and/or anxiety in mental health settings and may therefore have the potential to improve health in the CR population.[5 11 12] There are no existing published economic evaluations of MCT.

The PATHWAY Group MCT multicentre, single-blind, randomised controlled trial aims to evaluate the effectiveness and cost-effectiveness of group-based metacognitive therapy for CR patients with elevated anxiety and/or depressive symptoms.[5] Details of the main trial study protocol are available elsewhere.[5] In brief, participants were recruited from patients with heart disease and referred for CR, in five UK National Health Service (NHS) Trusts. Trial participants had to meet CR eligibility criteria, score 8 or more on either the anxiety and/or depression subscale of the Hospital Anxiety and Depression Scale (HADS), be aged 18 years old or more and able to read, understand and complete questionnaires in English. Patients who gave informed consent to participate in the trial completed baseline assessments and were randomised via telephone link to Manchester Clinical Trials Unit. Participants were allocated to the intervention/comparator arms in a 1:1 ratio via a minimisation algorithm (incorporating a random component) to maximise balance between the two arms on gender, HADS score and hospital site. Participants randomised to the intervention arm received 6 weekly sessions of group-based metacognitive therapy delivered by either CR professionals or research nurses. The intervention and control groups were both offered in the usual CR programme within their Trust. The primary outcome is

severity of anxiety and depressive symptoms at 4-month follow-up measured by the HADS total score.[13] Secondary outcomes are severity of anxiety/depression at 12-month follow-up, health status, severity of post-traumatic stress symptoms, strength of metacognitive beliefs and service use at 4-month and 12-month follow-up. The PATHWAY Group MCT trial is part of a wider NIHR (National Institute for Health Research) programme grant for applied research, which began in August 2014 and with an end date of February 2021.

The aim of the economic evaluation component of the PATHWAY Group MCT trial is to estimate the cost-effectiveness of the addition of MCT intervention to CR for patients with elevated anxiety and/or depression symptoms, in a UK setting from a health and social care perspective. This will consist of a within trial analysis using patient-level data collected during baseline and follow-up study time points, and an economic model synthesising data from the trial and wider literature.

## METHODS AND ANALYSIS

The economic evaluation of group MCT in CR will comprise of two analyses; a within-trial economic analysis prospectively designed to assess cost-effectiveness for the duration of the trial (12 months) and an economic model to explore the potential longer-term cost-effectiveness of MCT in this population and in different populations and settings.

Both analyses will use a cost-effectiveness acceptability analysis to assess the incremental cost-effectiveness of the addition of MCT intervention to the CR for patients with elevated anxiety and/or depression symptoms. The intervention is group-based MCT plus usual CR, and the comparator is usual CR alone. The perspective for the primary analysis (trial and model) is health (NHS) and social care service providers (costs) and CR participants (health benefits), which is the perspective recommended by National Institute for Health and Care Excellence (NICE).[14] Note that social care in the UK is delivered by a range of providers (public sector, commercial and voluntary) and encompasses a range of social support services.[15] The work will conform to the NICE recommended methods and will be reported in accordance with the Consolidated Health Economic Evaluation Reporting (CHEERS) statement.[14 16]

### Patient and public involvement

The patient and public involvement (PPI) group from the group MCT trial will be presented with the findings of the economic evaluation and consulted about possible dissemination activities that target patient groups.

### Within-trial analysis

The analysis will use individual patient-level service use and health benefit data from all participants recruited and allocated to a management arm in the PATHWAY Group MCT trial (n=332). The trial opened for recruitment in

July 2015 and closed in January 2018, follow-up finished in February 2019. The economic analysis of trial data will begin in April 2020 and will complete by September 2020. The analysis will compare the addition of group-based MCT to usual CR versus usual CR alone (standard practice) for individuals with elevated symptoms of depression and/or anxiety. The study sample for the economic evaluation is all participants randomised to receive the group-based MCT plus usual CR, or usual CR only. In brief, participants were people eligible and offered (referred to) CR as per NHS trust protocol, with a HADS score of 8 or more on the anxiety and/or depression subscale. The intervention, usual care and participant inclusion and exclusion criteria are summarised and described in detail in the trial protocol.[5] Usual CR (dependant on NHS site) includes varying degrees, psychological interventions to address distress such as relaxation, stress management and some cognitive challenging.

The time horizon of the within-trial primary cost-effectiveness analysis will be 12 months, to incorporate sufficient time for any impact of MCT on service use, subsequent costs and health benefit. This differs to the primary clinical effectiveness time horizon, which will be 4 months (note, a sensitivity analysis will look at cost-effectiveness at 4-month follow-up).[5]

### Outcomes

The measure of health benefit for the primary analysis will be the quality-adjusted life-year (QALY). This will be estimated from the EQ-5D-5L health status measure completed at baseline and 4-month and 12-month follow-up. The EQ-5D is a validated, generic health status measure, allowing for the comparison of health benefits across different disease areas. The EQ-5D has been shown to be a valid and responsive measure of health in CR patients.[17] Furthermore, the QALY and the EQ-5D are the measures recommended for economic evaluations by NICE.[18]

The EQ-5D-5L captures the following five domains of health status: mobility, self-care, usual activity, pain/distress and anxiety/depression. Each domain is rated on a 5-point scale of levels: no problems, slight problems, some problems, severe problems or unable to do activity. Resulting health status profiles will be converted to utility values using the published utility tariffs and methods recommended by NICE at the time of analysis. An EQ-5D-5L valuation set is available that reflects the preferences of members of the public in England for health states defined by the EQ-5D-5L.[19] However, concerns have been raised during the quality assurance of this value set and currently, NICE recommend that the crosswalk algorithm used to map from the EQ-5D-3L to the EQ-5D-5L continues to be used.[20] The economic evaluation will use the value set recommended by NICE at the time of analysis. The use of alternative value sets will be investigated in the sensitivity analysis.

Total QALYs will be estimated as follows:

**Table 1** Economic evaluation service use measures

| Service type | Unit measure |
| --- | --- |
| **NHS healthcare use (collected by a self-report economic patient questionnaire)** | |
| Hospital inpatient | Days per stay |
| Hospital day | Number of visits |
| Hospital outpatient | Number of visits |
| Accident and emergency | Number of visits |
| Primary care* | Number of visits |
| Community care† | Number of visits |
| **Metacognitive therapy (collected by the trial team)** | |
| Staff time (cardiac rehabilitation nurse, cardiac nurse, physiotherapist or occupational therapist) | Number of hours |
| Paper manual | By unit |
| CD exercise | By unit |
| **Cardiac rehabilitation attendance (collected by the trial team)** | |
| Exercise cardiac rehabilitation | Number of attendances |
| Education cardiac rehabilitation | Number of attendances |

*Examples include general practitioner.
†Examples include community-based mental healthcare and social support.
CD, compact disk; NHS, National Health Service.

$$QALY = \sum \left( \left[ U_i + U_{i+1} \right] / 2 \right) \times (t_{i+1} - t_i)$$

Here, U=utility value and t=time between assessments. The time between assessments is the time from baseline data collection to follow-up.

### Resource use and costs

Direct costs will be estimated for the primary analysis. Service use included in the economic evaluation is listed in table 1. With the exception of the intervention and usual care cardiac rehabilitation, data on the healthcare resources used for each participant were collected from an economic patient questionnaire at baseline and follow-up (4 and 12 months). The questionnaire was adapted from surveys used by the authors in previous trials.[21–24] The changes to the questionnaire were developed with the service user group and the clinical experts in the research team. The questionnaire was pilot tested before use. At baseline this is completed by the participant with assistance from a researcher. This means participants have experience of completing it once with help before they need to complete it unassisted. At follow-up participants complete the questionnaire on their own but were aware a researcher could help them complete the questionnaire if needed. A copy of the questionnaire is provided in the online supplementary material and has been uploaded to the Database of Instruments for Resource Use Measurement (DIRUM).[25] The questionnaire asks patients to report service use for the follow-up period (time since the last questionnaire or 3 months at

baseline) and for the last month. If a participant struggles to recall service use for the full follow-up but knows the last month, this is included to help prompt the participant. For the primary analysis, participants with 1-month but not the full assessment period service use data will be treated as having missing data. The missing data will be imputed using the multiple imputation process defined for all participants with missing data. Two sensitivity analyses will explore the impact of using the 1-month recall data (see table 2).

Note that from the reported service use, psychological treatment outside of cardiac rehabilitation and MCT will also be costed. MCT attendance data collected during the trial by the research team will be used to estimate a per participant cost for the MCT intervention. In the primary analysis, this will include staff costs and the cost of materials. The average group size from the trial will be used to calculate a cost per session per participant, which will be multiplied by the number of sessions attended by each participant. Staff time to deliver MCT and attendances at cardiac rehabilitation will be collected by the research team.

The unit costs of NHS and social care services will be derived from national average unit cost data. These unit costs are published annually in the NHS reference costs database, and in the Unit Costs of Health and Social Care document published by the Personal Social Services Research Unit, University of Kent. The price year for all costs will reflect the most recent unit costs at the time of analysis. The total direct costs of service use for each trial arm (including MCT and cardiac rehabilitation) will be estimated by summing the costs of each resource by the reported use to provide health and social care.

### Missing data

Analysis of the economic data will use an intention-to-treat approach that includes service use, cost and health benefit data for all services users, regardless of whether they completed the planned follow-up and all the measures used at assessment. It is highly likely that data will be missing, either from loss to follow-up or incomplete data collection.

Single imputation will be used to impute missing baseline data. This simple approach is appropriate for baseline data, although in reality this is unlikely to have a significant impact as participants who have missing data at baseline are less likely to have follow-up data.[26 27] Baseline variables are observed at entry to the trial and vary by the individual rather than random allocation to trial arms or values at follow-up. In contrast follow-up values for clinical and economic measures may be dependent on trial allocation and baseline. Accordingly, single imputation for baseline measures of cost, utility and clinical indicators ensures the estimated values are independent of treatment allocation and follow-up values. Single imputation is likely to be more efficient than multiple imputation. Multiple imputation (MI) will be used to impute missing follow-up data for costs and QALYs, which

assumes that the data are missing at random (conditional on observed responses to clinical and economic measures and baseline covariates); MI of both costs and QALYs is recognised as an appropriate approach to deal with missing observation and missing follow-up data.[26] Missing values will be imputed for each follow-up point. All missing follow-up cost and utility data will be assumed to be missing at random. To make best use of available data, we plan to impute costs by category (inpatient, outpatient and primary/community care) and utility by individual EQ-5D domain (mobility, self-care, usual activities, pain/discomfort and anxiety/depression). A balance is required between including all items and the stability of the imputation model if there is a high level of missing data. The stability of the model reduces as the number of variables with missing data increases. If feasible, all individual items in each cost category will be included in the imputation model. Total costs and QALYs (baseline to follow-up) will be generated using Stata passive estimation commands. The final imputation plan will be refined following a review of missing data (exploration of whether there are any patterns of missingness) and according to the patterns of service use and EQ-5D data. Two sensitivity analyses will explore whether the results vary by approach taken to missing data. The first will use complete case analysis. The second will use a pattern mixture approach that assumes compliance and clinical outcomes are systematically worse in people with missing or with incomplete follow-up.[26 28] The specification of the analysis will be finalised after review of the complete and missing data patterns and discussion with clinical experts about plausible assumptions. If appropriate this analysis will identify the threshold values of compliance and clinical outcome that would lead to a change in conclusions about the relative cost-effectiveness of the within trial economic analysis.

Multiple imputations will be conducted in Stata V.15, using predictive mean matching and sequential chained equations. Regression models used to impute missing data will be based on key covariates associated with costs or health benefits. A long list of potential covariates was identified from a recent systematic review of economic evaluations in the population undergoing cardiac rehabilitation and the baseline data collected within this randomised controlled trial (see online supplementary material). Discussions with clinical experts in the team will be used to assess the relevance and logic of using these variables as covariates in our analysis. This approach will be supplemented with pooled descriptive analyses and regression analyses to identify the final set of covariates.[6] A list of potential covariates is included in the online supplementary material.

### Primary analysis

Costs and health benefit for the primary and sensitivity analyses will be estimated from baseline to end of 12-month follow-up, to estimate the incremental cost-effectiveness of the addition of MCT intervention.

**Table 2** Within-trial sensitivity analysis

| Assumptions/variables | Changes | Rationale |
|---|---|---|
| Missing data are assumed to be missing at random | ▶ Complete case analysis<br>▶ Pattern mixture approach | The complete case analysis will use only the observed data and will provide insight to the result for the group of participants with complete follow-up and complete data (evaluable cohort).<br>The pattern mixture approach will assess the impact of assuming that compliance and clinical outcomes are systematically worse in people with missing or with incomplete follow-up. It will be assumed that these differences will also lead to systematic worse health benefits and higher costs for participants with missing data[26 28]<br>The results of the complete case and pattern mixture analyses will be compared with the primary analysis based on multiple imputed data sets to give an indication of how robust the cost-effectiveness estimate is to assumptions about missing data. |
| Use of 1-month recall data | ▶ Assume that service use is equally distributed over the assessment period and that the 1-month data can be multiplied up to 4-month or 8-month follow-up.<br>▶ Add the 1-month cost estimates to the multiple imputation process, depending on the pattern of missing data and sufficient participants with both 1-month and 4-month or 8-month cost estimates. | The primary analysis treats the 1-month service use data as uninformative for participants who are missing data for the assessment period. These two sensitivity analyses explore the impact on the results of this approach. |
| Treatment received rather than intention-to-treat | ▶ The MCT group will only include those who attended at least one MCT session | The primary analysis will be an intention-to-treat approach. Recognising that not all patients assigned to the MCT intervention will attend, we will also run an analysis dividing the group using the recorded MCT attendance data. |
| Subgroup analyses | ▶ History of anxiety and depression at baseline<br>▶ Gender<br>▶ HADS at baseline (severity of depression and anxiety) | Subgroup analysis will be conducted if there are enough numbers of participants in each group, to explore how the likely cost-effectiveness differs according to the population. Note, this is highly explorative and will be used to guide the economic model and generate further research questions rather than producing definitive results/conclusions. |
| Measure of benefit | ▶ Hospital Anxiety and Depression Scale (HADS)<br>▶ Cases of anxiety and/or depression<br>▶ Reliable improvement in HADS score<br>▶ Return to productive activity | Secondary analyses will explore the cost-effectiveness of MCT intervention using a problem-specific measure of effectiveness, rather than the generic QALY. This will look at the cost per point change in the HADS, as well as cost per reliable improvement in HADS and cost per case of anxiety and/or depression avoided. Cases of anxiety and depression will be classified using the HADS score (8 to 10=mild, 11 to 14=moderate and 15 to 21=severe).[45] In our analysis we will focus on the cut-off used in the inclusion criteria for the Group MCT trial (≥8 on either the depression or anxiety subscale of the HADS). Productive activity (employed, voluntary work, in education or training, carer or looking after the family or home). These alternative measures of benefit will be used to aid decision-makers who may be less familiar with QALYs or have different objectives/targets. |
| Utility value set to estimate QALYs | ▶ Alternative EQ-5D value sets (Devlin et al 2018[46]; van Hout et al, 2012[20]) | Secondary analyses will explore the impact of using alternative value sets to calculate QALYs. The primary analysis will use the value set recommended by NICE at the time of the analysis, the remaining EQ-5D value set will be used in a sensitivity analysis. This will assess the impact of the different methods that can be used to estimate utility. |
| Cost of MCT intervention | ▶ Assumed larger group size<br>▶ Inclusion of wider costs (training and catering) | Sensitivity analysis will be conducted in which a larger average group size is assumed because if MCT was implemented in CR, a larger group size would be likely. Additionally, separately to group size, we will also look at a more comprehensive (although uncertain) method of costing the MCT intervention, which will include staff training costs |

| Table 2 | Continued | |
| --- | --- | --- |
| Assumptions/variables | Changes | Rationale |
| Time horizon | ► 4 months | The final trial time follow-up is 12 months. A secondary analysis will consider the 4-month follow-up (the primary follow-up of the trial), to assess the impact of different follow-up periods on cost-effectiveness results. |

Adherence to CR has been defined by the study team as the attendance to four or more sessions to each component of usual CR (exercise sessions and educational talks).
CR, cardiac rehabilitation; MCT, metacognitive therapy; QALY, quality-adjusted life-year.

The primary measure of interest for the economic analysis is the incremental cost-effectiveness ratio (ICER). Rather than considering cost and health benefit separately, the ICER is a joint measure of both. It is calculated by dividing the difference in costs (net costs) by the difference in QALYs (net QALYs) between any two interventions. The ICER represents the additional cost of an intervention per additional QALY gained:

$$\text{ICER} = \frac{\text{Cost}_{\text{MCT intervention plus usual care}} - \text{Cost}_{\text{usual care}}}{\text{QALYs}_{\text{MCT intervention plus usual care}} - \text{QALYs}_{\text{usual care}}}$$

Note that if the intervention is cost saving and produces more QALYs when compared with usual care (ie, it falls in the South East quadrant of the cost-effectiveness plane), an ICER will not be presented as intervention is dominant in such a scenario. Likewise, if intervention is dominated (ICER lies in the North West quadrant of the cost-effectiveness plane) it will be described in this way rather than calculated and presented numerically.

Descriptive analysis and data manipulation will be conducted using SPSS V.25 and the main statistical analyses will be conducted in Stata V.15.

Regression analysis will be used to estimate the net costs and QALYs of MCT. Cost and QALY data may not be normally distributed. Graphical summaries and descriptive statistics will be used to assess the distribution of the pooled cost and QALY data and identify the appropriate regression models for the analysis (eg, generalised linear model with gamma, log distribution for costs). Key covariates will be included in the regression models to control for baseline factors that may influence costs or QALYs. The covariates for these analyses will be identified using the approach outlined for the multiple imputation described in the previous section.

The ICER measures the cost per QALY gained by an intervention which then raises the question of whether the additional cost of a QALY is worth paying for. To help address this, the ICER can be compared with benchmark or threshold values of how much decision-makers may be willing to pay to gain one additional QALY. This is analogous to placing a monetary value on one QALY. This then allows cost-effectiveness acceptability analysis, which is recommended by NICE for health technology appraisals.[14]

In the UK there is no universally agreed cost-effectiveness threshold value. One commonly reported threshold is from NICE in England of approximately £20 000 to £30 000 per QALY.[29] However, there is a lack of consensus around the appropriate threshold and some argue that it has reduced as expenditure has been constrained.[30–32] Therefore, the monetary value of simulated QALYs will be varied from £0 to £30 000 to reflect a range of hypothetical willingness to pay thresholds (WTPT).

### Handling uncertainty

The estimates of costs and health benefits from the regression analyses will be bootstrapped to simulate 10 000 pairs of incremental cost and QALY outcomes of the MCT intervention.[14] This captures the relationship between costs and QALYs. The pairs of net costs and QALYs will be plotted on a cost effectiveness plane to illustrate the level of uncertainty in the data.

Each of the net QALY estimates from bootstrap simulation results can be revalued by multiplying it by a willingness to pay threshold. Using these revalued QALY estimates it is then possible to estimate the net benefit statistic (NB) for each pair of simulated net costs and net benefits as:

NB = (O × threshold) – C, where O = net outcome score and C = net cost.

This process will be repeated for the WTPT values of interest to generate a cost effectiveness acceptability curve.

Additional sensitivity analysis will be used to test the impact of the study design on the ICER and results of the cost-effectiveness acceptability analysis. This will include subgroup analysis to estimate the impact of heterogeneity across the population, and scenario analysis to estimate the impact of study design on the cost-effectiveness of intervention. A detailed description of the likely sensitivity analysis and rationale are described in table 2. Note that the sensitivity analysis is informed by data collected in the trial.

### Long-term health economic model

A de novo economic model will be constructed using Microsoft Excel and programmed in Visual Basic for Applications. There are two key reasons for the development of this model: (1) to explore the cost-effectiveness of MCT over a longer time horizon, and (2) to explore the cost-effectiveness of MCT in different populations and settings. The model design work will begin in April 2020, targeted literature reviews to identify relevant data

will be performed in April and June 2020, with the model expected to be constructed and validated by September 2020.

### Model design and validation

The economic model structure will be developed from the trial care pathways and literature reviews of existing models for depression interventions in patients with and without comorbidities. Iterative, structured discussion with the wider trial team which includes clinical advisors for the project (including cardiac specialists and psychologists) will be used to assess the face validity of the structure and identify necessary amendments.

Once the model structure is drafted it will be discussed with an external Health Economics Adviser (part of the Trial Steering Committee) and the trial PPI group to assess whether it captures the key events and outcomes. An initial model structure has been drafted and will be used as a basis for discussion but may not reflect the final chosen model structure. The model structure will also be validated by using the trial data and follow-up data and assessing whether the results are within ±5% of the within trial economic analysis for the trial time horizon.

The literature discusses the many benefits of complex models, such as microsimulation models, which can incorporate patient heterogeneity and history, and can more closely reflect clinical pathways and time frames, such models are data hungry and computationally burdensome. Given that there is limited evidence (eg, over a long time frame) for MCT in CR, for the potential patient population (eg, history leading up to a cardiac event) and the population is likely to be heterogeneous, many assumptions would need to be made to apply a discrete event simulation methodology, likely leading to an 'Occam's razor' scenario. A recent review found that Markov models are more common than other modelling methods when looking at mental health interventions.[33] Another recent review looking at model-based analyses specifically of treatments for depression found that the majority of papers used decision trees (21/41) or Markov models (15/41).[34] Therefore, a simpler model design (decision tree and/or Markov) is likely to be preferred.

### Identifying data sources

Data from the trial will likely inform the majority of short-term parameters within the base case economic model. This will be supplemented with structured literature reviews to identify parameters and alternative inputs for the short-term that might help to inform analyses in alternative settings and population. Examples of inputs that will be identified will include uptake rates in cardiac rehabilitation, utilities, costs, mortality and likelihood of depression/anxiety relapse and remission. Evidence relevant to the cardiac population will be prioritised. Parameter tables will be drafted and shared with the clinical advisers on the project to discuss and assess clinical validity prior to use in the model. Data uncertainty will be explored in the sensitivity analysis outlined below.

### Primary analysis

The model will report the estimated costs, QALYs and ICER for MCT plus usual care, versus usual care, with a 5-year time horizon. Costs and outcomes will be discounted at a rate of 3.5% in line with UK guidelines.[14] Probabilistic sensitivity analysis (PSA) will be performed, using Monte Carlo simulation with 10 000 iteration runs. The Monte Carlo simulation samples from the distribution of possible values for each parameter in the decision model so that mean costs and outcomes, and measures of variance (SD and 95$^{th}$ percentiles) can be estimated to assess the uncertainty inherent in the data used for the model. PSA results will be presented on a cost-effectiveness plane and cost-effectiveness acceptability curves will be produced, in line with NICE recommendations for health technology appraisals.[14]

### Sensitivity analysis

Like the trial analysis, sensitivity analysis will be used to explore the impact of design choices on cost-effectiveness, as well as to consider potential longer-term outcomes. A detailed description of the likely sensitivity analysis and rationale are described in the online supplementary material. One of the key sensitivity analyses will be the investigation of the impact of different assumed durations of MCT effect. Details of the analysis will be refined following analysis of trial data and the review of wider literature and will be validated by clinicians. Note that the results for all sensitivity analysis will be run and reported in the same way as the primary analysis. Additionally, the model will also include one-way sensitivity analysis, varying parameters between their lower and upper bounds and presenting the results of this in a Tornado diagram.

## DISCUSSION

This economic evaluation will comprehensively assess the potential cost-effectiveness of MCT for CR patients. Work will start with the economic evaluation integrated into the randomised controlled trial, prospectively designed to collect resource use and health outcome data in the trial participants. A robust within-trial economic evaluation will be conducted with a time horizon of 12-months. Pre-specified subgroup and sensitivity analyses will test the impact of design choices on the results and conclusions of the primary analysis. The economic evaluation work will be expanded with the design and construction of an economic model, intended to estimate the potential cost-effectiveness of MCT for CR over a longer time horizon, as well as investigating results across different settings and populations. To the team's knowledge this will be the first economic evaluation of MCT. It will also add to the very limited economic evidence for psychological therapies within CR.[6]

The work will be published in peer reviewed journals and presented at economic and/or clinical focussed conferences. The PPI group from the group MCT trial will be presented with the findings of the economic evaluation

and consulted about possible dissemination activities that target patient groups. Published work will include discussion on the limitations, generalisability of the evidence to setting outside of the UK and a comparison to the exiting wider evidence for the cost-effectiveness of CR. The latter will be identified from existing systematic reviews and more recently published work.[6 35–39]

There are a number of anticipated challenges of the work that may result in limitations associated with the economic evaluations. Regarding the within-trial analysis missing data poses a threat to economic evaluations. However, imputation methods are planned to reduce the impact of this and a complete-case analysis will be conducted as part of the scenario analysis.[40] In addition, the final publication will report the potential impact of missing data, for example, summarising the level of missingness by data type. Nevertheless, the multiple imputation approach assumes that the data are missing at random (conditional on observed responses to clinical and economic measures and baseline covariates), which may not be valid. A high level of complete data on the primary clinical outcome, is anticipated, which may help to strengthen analysis based on the missing at random assumption. However, it may not be sufficient to account for systematic differences in costs and health benefits between participants with and without missing data or incomplete follow-up. Sensitivity analysis using (i) complete case analysis and (ii) a pattern mixture approach, which assumes compliance and clinical outcomes are systematically worse in people with missing or with incomplete follow-up.[26 28]

Healthcare resource use will be self-reported by participants in the trial. The literature notes that self-report questionnaires are a valid approach to collecting healthcare resource use[41] and were used in previous UK cost-effectiveness analyses of cardiac rehabilitation.[42 43] However, self-report forms are susceptible to recall bias and missing data. To improve feasibility and reduce bias, the service use questionnaire was adapted from those used in previous trials (references) and changes made jointly with service users and clinical experts. The forms were pilot-tested and refined. Service users complete the form at baseline by interview, to familiarise them with the forms. At follow-up, participants are also offered telephone assistance with completing the forms. Nevertheless, reliance on self-reported service use to estimate costs is a limitation of the economic analysis. It is not clear whether this will lead to systematic under or over reporting of costs and/or contribute to non-systematic inaccuracy of reported service use and costs.

Collecting some or all of the service use data from routinely collected electronic data sets in England was explored. Issues encountered included the need for secondary, primary, community and social care data for the study participants. Linked electronic data across these sectors is not typically available in England at local or national level. Ethical and governance approval and funding constraints meant it was beyond the scope of this study to identify all services used by participants, and

review the case notes held by each service the participants used. Centrally collected databases in England, such as NHS Digital, typically do not include linked data for the range of services included in the study. This meant that the majority of service use data would need to be collected directly from participants.[36] Access to hospital service use data for named patients via NHS Digital was paused and not available when the funding application was made. At the point at which the data collection methods needed to be finalised, the time to access data from NHS Digital was estimated at 12 months from the last participant completing the final scheduled follow-up to ensure the records were complete. Accordingly, while collecting service use data from participants is a limitation of the economic analysis, sourcing some or all of this data was not feasible within the funding and time constraints for this study.

The trial is being conducted in the North West of England; cardiac services and populations vary across the country, therefore there may be generalisability issues. In particular, the types of people offered CR is very broad and in some settings certain groups may be prioritised and attendance may vary.[1 2] If possible, the economic modelling work will explore the potential cost-effectiveness of MCT in other settings and populations in an attempt to understand how this may vary. However, the variation across settings and populations is likely to reduce generalisability although decision-makers should consider whether the intervention and participants are reflective of their setting. As noted in the introduction, only around half of those offered CR in the UK attend and there is insufficient data available about the proportion specific to those with symptoms of anxiety and/or depression.[1] This analysis is limited to people who attend cardiac rehabilitation. Important questions remain around how to increase attendance and what methods would be cost-effective in doing so, which are outside of the scope of this analysis but should be addressed by future studies.

Multimorbidity is a complicating factor of the analysis and although we will attempt to understand and discuss how physical and mental health interact in this population the data is likely to be limited, and it is likely that the model will focus on depression and anxiety. The development of the economic model will require data from the wider published literature, but until searches are conducted it is not known how abundant and robust this data will be. Given the complexity of economic evaluation and the numerous possibilities for the economic modelling work, practical steps, such as reviewing the existing literature and discussions with clinical experts, will aim to ensure that the economic evaluations are as meaningful and useful in supporting decision-making as possible.

The discussed study will provide evidence on the potential cost-effectiveness of MCT in CR in the UK. This is highly relevant for decision-makers given the prevalence of depression and/or anxiety in the CR population, the economic and health burden associated with both cardiac

problems and depression and/or anxiety and calls for more integration of physical and mental health services.[44]

**Author affiliations**
¹Manchester Centre for Health Economics, The University of Manchester, Manchester, UK
²Faculty of Biology, Medicine and Health, School of Psychological Sciences, Manchester Academic Health Science Centre, The University of Manchester, Manchester, UK
³Research & Innovation, Greater Manchester Mental Health NHS Foundation Trust, Manchester Academic Health Science Centre, Manchester, UK
⁴Department of Health Sciences, University of York, York, UK
⁵Centre for Primary Care, The University of Manchester, Manchester, UK
⁶Institute of Cardiovascular Sciences, The University of Manchester, Manchester, UK

**Contributors** GES, AW and LMD formulated the research questions. GES wrote the first draft of the protocol. DB, AW, PJD, TH, LC, DR and LMD contributed to the final writing of the paper, AW is the chief investigator.

**Funding** This paper presents independent research funded by the National Institute for Health Research (NIHR) under its Programme Grants for Applied Research (PGfAR) Programme (Grant Reference Number RP-PG-1211-20011). The views expressed are those of the authors and not necessarily those of the NIHR or the Department of Health.

**Competing interests** None declared.

**Patient and public involvement** Patients and/or the public were involved in the design, or conduct, or reporting, or dissemination plans of this research. Refer to the Methods section for further details.

**Patient consent for publication** Not required.

**Provenance and peer review** Not commissioned; externally peer reviewed.

**ORCID iDs**
Gemma E Shields http://orcid.org/0000-0003-4869-7524
Adrian Wells http://orcid.org/0000-0001-7713-1592
Patrick Doherty http://orcid.org/0000-0002-1887-0237
David Reeves http://orcid.org/0000-0001-6377-6859
Lora Capobianco http://orcid.org/0000-0001-6877-8650
Anthony Heagerty http://orcid.org/0000-0002-9043-2119
Linda M Davies http://orcid.org/0000-0001-8801-3559

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
