## [Reviewer comments · BMJ Open]

ARTICLE DETAILS

TITLE (PROVISIONAL)	Protocol for the economic evaluation of metacognitive therapy for cardiac rehabilitation participants with symptoms of anxiety and/or depression
AUTHORS	Shields, Gemma; Wells, Adrian; Doherty, Patrick Joseph; Reeves, David; Capobianco, Lora; Heagerty, Anthony; Buck, Deborah; Davies, Linda

VERSION 1 – REVIEW

REVIEWER	Dr. M.E. van den Akker-van Marle Department of Biomedical Data Sciences, unit Medical Decision Making, Leiden University Medical Center, The Netherlands
REVIEW RETURNED	13-Dec-2019

GENERAL COMMENTS	This protocol for the economic evaluation of metacognitive therapy for cardiac rehabilitation participants, is not very exciting as is normal for a protocol paper, but it is written accurately and in detail. Some comments: - The dates of the study should included in the manuscript. Introduction - An important problem of cardiac rehabilitation, at least in our country, is that patients do not start cardiac rehabilitation. Please indicate in the introduction which part of the patients in the UK that are eligible for rehabilitation, does participate. Methods and analysis: - What is the definition of social care, and what is the difference with health care? This might be not familiar to non-UK readers. E.g. I would expect a physiotherapist to be health care, but in the supplementary materials it is included in the social support services.- How many patients will be included?- At what moments in the follow up the EQ 5D will be completed? Only at 4 and 12 months?- Is the data on healthcare use collected by a questionnaire or an interview with the patient? Please add to the manuscript.- Why is both a one-month and three-month recall period used? Which answer is used in the calculations? How are these answers translated into annual costs?- Why is single imputation used to impute missing baseline data and multiple imputation for missing follow up data? Please explain in the manuscript.
--

	- Why are costs imputed by category and not by item? What if only one item is missing in a category for a patient, is the entire category missing by then (and data on the other items lost)? - The ICER should only be calculated if both the incremental costs and the incremental effects are negative or positive, otherwise one of the interventions is dominant.
--	--

REVIEWER	Marie Kruse University of Southern Denmark Denmark
REVIEW RETURNED	04-Feb-2020

GENERAL COMMENTS	Review of: Protocol for the economic evaluation of metacognitive therapy for cardiac rehabilitation participants with symptoms of anxiety and/or depression The study is a within-trial economic evaluation of metacognitive therapy for cardiac patients with anxiety and/or depression. It is well-written and well structured, and applies state-of-the art methodology throughout. I have a few minor reservations and one perhaps a bit more important. Both groups are eligible for cardiac rehabilitation and the control group receive cardiac rehabilitation without the metacognitive therapy. Both groups score 8 or above on the HADS-scale, which means that also the control group complies with the definition of being clinically depressed or anxious. However, no psychological treatment is offered to this group. Therefore, it is likely that control patients to some extent will seek treatment outside the trial setting, privately or in other parts of the health care sector. While this element of treatment as usual is difficult to measure, it represents a bias in the direction of the intervention seeming more costly and less effective. If possible, patients should report their use of services outside the NHS. It is a bit unclear if participants are CR participants or merely found eligible for CR. There is an equity aspect of CR, that is important to address when discussing external validity. Re: external validity, the authors state that the analysis will be relevant for the UK, while I would argue that it could indeed be relevant in other countries as well. I agree, that a four-month time horizon is way too short to capture the effect of the therapy, I am not sure that a 12-month time window is sufficient. The long-term analysis (model-based) should incorporate a discussion of how benefits will develop if the four-month sensitivity analysis shows a large increase and the 12-month follow-up shows a fall? What is known about the development of HRQoL and mental health over time? My most important reservation regards the measurement of resource use: if (some of the) cost data is collected from patients, how can the analyses be conducted on intention-to-treat basis? There will most likely be an attrition bias, where those lost to follow-up have higher health care costs than compliant patients. This fact seems to be disregarded when the description of the imputation states that missing values are assumed to missing at random. I would like to challenge this assumption.
---

	In addition, I believe HRG's are much more precise in terms of resource use, than patient reported length of stay etc. What is the rationale of not using HRG's/patient records? The intervention and CR costs are collected by the health professionals however I can't read the details of this data collection anywhere. E.g. table 1 reads 'number of hours' of health professionals' time, however what if they only spend 15 minutes? Would that be disregarded, and where do the unit costs from these hours come from? The sensitivity analysis is well described, and comprehensive. One of the items is: using cases of anxiety and/or depression as measure of benefit. How is this information obtained? Self-reported or patient records? In case of the former, there must be some information bias? In case of the latter, if patient records are accessible, I'd recommend using them for cost analysis, cf above.
--	--

VERSION 1 – AUTHOR RESPONSE

Comment	Action to address
Reviewer 1	
This protocol for the economic evaluation of metacognitive therapy for cardiac rehabilitation participants, is not very exciting as is normal for a protocol paper, but it is written accurately and in detail.	Thank you, we appreciate that protocols do not make interesting pieces of work to review and really appreciate you taking your time to do this.
- The dates of the study should included in the manuscript.	Dates have been added throughout, including: The PATHWAY Group MCT trial is part of a wider NIHR programme grant for applied research, which began in August 2014 and with an end date of February 2021. The trial opened for recruitment in July 2015 and closed in January 2018, follow-up finished in February 2019. The economic analysis of trial data will begin in April 2020 and will complete by September 2020. The model design work will begin in April 2020, targeted literature reviews to identify relevant data will be performed in April and June 2020, with the model expected to be constructed and validated by September 2020.
- An important problem of cardiac rehabilitation, at least in our country, is that patients do not start cardiac rehabilitation. Please indicate in the introduction which part of the patients in the UK that are eligible for rehabilitation, does participate.	Further detail has been added into the introduction and discussion to highlight this. This includes information on who is eligible to attend, the key groups actually attended and the percentage of people offered CR who actually attend. This detail includes: The population offered CR is variable; the British Association for Cardiovascular

	Prevention and Rehabilitation propose three groups of priority, including acute coronary syndrome, coronary revascularisation and/or heart failure, with further groups who should be offered CR if possible (including stable angina amongst others) [2]. The greatest number of attendees of CR in the UK come from populations with myocardial infarction, percutaneous coronary intervention, coronary artery bypass graft and heart failure [1]. And Around 50% of those offered CR attend [1]. In addition, in the limitations the following text has been added: In particular, the types of people offered CR is very broad and in some settings certain groups may be prioritised and attendance may vary [1,2]. And: However, the variation across settings and populations is likely to reduce generalisability although decision makers should consider whether the intervention and participants are reflective of their setting.
- What is the definition of social care, and what is the difference with health care? This might be not familiar to non-UK readers. E.g. I would expect a physiotherapist to be health care, but in the supplementary materials it is included in the social support services.	There is no exact definition of social care in the UK, it relates more to who delivers services and where they take place. Text has been added to clarify this: Note that social care in the UK is delivered by a range of providers (public sector, commercial and voluntary) and encompasses a range of social support services [15].
- How many patients will be included?	This has been added to the first line of the within-trial analysis which now reads: The analysis will use individual patient-level service use and health benefit data from all participants recruited and allocated to a management arm in the PATHWAY Group MCT trial (n=332).
- At what moments in the follow up the EQ 5D will be completed? Only at 4 and 12 months?	It will be collected at baseline and follow-up. We have revised the text to read: The EQ-5D-5L health status measure completed at baseline and 4 and 12-month follow-up.
- Is the data on healthcare use collected by a questionnaire or an interview with the patient? Please add to the manuscript.	The following text has been added to the manuscript to clarify this: At baseline this was completed by the patient with assistance from a researcher and at follow-up it was completed alone by the patient, except for a minority of follow-ups that were performed over the phone with the assistance of a researcher.

- Why is both a one-month and three-month recall period used? Which answer is used in the calculations? How are these answers translated into annual costs?	The following text has been added to expand on this: The questionnaire asks patients to report service use for the follow-up period and for the last month. If a participant struggles to recall service use for the full follow-up but knows the last month, this means that there is at least some data on service use. If this is a routine service, the one-month response will be multiplied up to the full follow-up period (only if the full follow-up period response is missing). Note, this is aligned with service use collection tools that have been successfully used by the research team on previous trials.
- Why is single imputation used to impute missing baseline data and multiple imputation for missing follow up data? Please explain in the manuscript.	Further text has been added to expand on imputation and to address this comment: Single imputation will be used to impute missing baseline data. This simple approach is appropriate for baseline data, although in reality this is unlikely to have a significant impact as participants who have missing data at baseline are less likely to have follow-up data [22,23]. Baseline variables are observed at entry to the trial and vary by the individual rather than random allocation to trial arms or values at follow up. In contrast follow up values for clinical and economic measures may be dependent on trial allocation and baseline. Accordingly, single imputation for baseline measures of cost, utility and clinical indicators ensures the estimated values are independent of treatment allocation and follow up values. Single imputation is likely to be more efficient than multiple imputation. Multiple imputation of follow up variables accounts for uncertainty in the missing data as well uncertainty in the estimated parameters included in the imputation model. MI of both costs and QALYs is recognised as an appropriate approach to deal with missing observation and missing follow-up data [21].
- Why are costs imputed by category and not by item? What if only one item is missing in a category for a patient, is the entire category missing by then (and data on the other items lost)?	Text has been added to clarify: To make best use of available data, we plan to impute costs by category (inpatient, outpatient and primary/community care) and utility by individual EQ-5D domain (mobility, self-care, usual activities, pain/discomfort and anxiety/depression). A balance is required between including all items and the stability of the imputation model if there is a high level of missing data. The stability of the model reduces as the number of variables with missing data

	increases. If feasible, all individual items in each cost category will be included in the imputation model.
- The ICER should only be calculated if both the incremental costs and the incremental effects are negative or positive, otherwise one of the interventions is dominant.	This is a good point. Text has been added to address this: Note that if the intervention is cost saving and produces more QALYs when compared to usual care (i.e. it falls in the South East quadrant of the cost-effectiveness plane), an ICER will not be presented as intervention is dominant in such a scenario. Likewise, if intervention is dominated (ICER lies in the North West quadrant of the cost-effectiveness plane) it will be described in this way rather than calculated and presented numerically.
Reviewer 2	
The study is a within-trial economic evaluation of metacognitive therapy for cardiac patients with anxiety and/or depression. It is well-written and well structured, and applies state-of-the art methodology throughout. I have a few minor reservations and one perhaps a bit more important.	Thank you for your considered responses and for taking the time to review our protocol.
Both groups are eligible for cardiac rehabilitation and the control group receive cardiac rehabilitation without the metacognitive therapy. Both groups score 8 or above on the HADS-scale, which means that also the control group complies with the definition of being clinically depressed or anxious. However, no psychological treatment is offered to this group. Therefore, it is likely that control patients to some extent will seek treatment outside the trial setting, privately or in other parts of the health care sector. While this element of treatment as usual is difficult to measure, it represents a bias in the direction of the intervention seeming more costly and less effective. If possible, patients should report their use of services outside the NHS.	Further text has been added to clarify that usual CR (accessed by intervention and control) may include some psychological intervention: Usual CR (dependant on NHS site) includes varying degrees, psychological interventions to address distress such as relaxation, stress management and some cognitive challenging. In addition, we have clarified that participation in the study does not preclude people undergoing other treatments and that data on additional psychological support will be available for participants in both groups: Note that from the reported service use, psychological treatment outside of cardiac rehabilitation and MCT will also be costed.
It is a bit unclear if participants are CR participants or merely found eligible for CR. There is an equity aspect of CR, that is important to address when discussing external validity. Re: external validity, the authors state that the analysis will be relevant for the UK, while I would argue that it could indeed be relevant in other countries as well.	More detail has been added to clarify this under the within-trial analysis header: In brief, participants were people eligible and offered (referred to) CR as per NHS trust protocol, with a HADS score of 8 or more on the anxiety and/or depression subscale.

	Note that the comments from reviewer 1 that have been addressed expand on the patient population attended CR in the UK and limitations according to generalisability.
I agree, that a four-month time horizon is way too short to capture the effect of the therapy, I am not sure that a 12-month time window is sufficient. The long-term analysis (model-based) should incorporate a discussion of how benefits will develop if the four-month sensitivity analysis shows a large increase and the 12-month follow-up shows a fall? What is known about the development of HRQoL and mental health over time?	The 4-month follow up is the primary time horizon for the clinical effect, as treatment (MCT and CR) will have completed in this time even if participants do not always attend the 8 weeks of rehab consecutively due to hospital appointments and illness. As it is a psychological therapy, if an effect does not occur by the end of treatment, it is highly unlikely it will over a longer time horizon. The longer follow-up (12-months) was chosen for the economic analysis to reflect the fact that changes in the key economic measures (service use and utility) are unlikely to happen immediately following symptoms changes. We have expanded on the consideration of the duration of effect in the supplementary table that outlines potential model sensitivity analysis. In the main body of the text we have highlighted: One of the key sensitivity analyses will be the investigation of the impact of different assumed durations of MCT effect. Details of the analysis will be refined following analysis of trial data and the review of wider literature and will be validated by clinicians. As outlined in the discussion: The development of the economic model will require data from the wider published literature, but until searches are conducted it is not known how abundant and robust this data will be. We have added an example here related to the reviewer comment: For example, data on the natural progression of depression and/or anxiety in this cohort.
My most important reservation regards the measurement of resource use: if (some of the) cost data is collected from patients, how can the analyses be conducted on intention-to-treat basis? There will most likely be an attrition bias, where those lost to follow-up have higher health care costs than compliant patients. This fact seems to be disregarded when the description of the imputation states that missing values are assumed to missing at random. I would like to challenge this assumption.	The analyses will be conducted on an intention to treat basis because participants will be analysed in the groups to which they were randomised. It is difficult to predict how attrition bias may impact, as it could also be the case that participants who experience health gains feel they have nothing to gain from being in the study and become lost to follow up. Either way this cannot be predicted. In addition, there are a

	number of mechanisms which can lead to higher usage of health care, for example as a result of the intervention appropriate engagement with healthcare services may increase whereby someone goes sooner to see their GP and has their problem resolved in primary care thus avoiding a more costly inpatient admission when symptoms worsen. As such there is unlikely to be an identifiable relationship between the propensity of a value to be missing and its value (i.e. missing not at random). However, before doing the multiple imputation we will explore patterns of missingness within the data which will inform our final approach. The text has been expanded to read: The final imputation plan will be refined following a review of missing data (exploration of whether there are any patterns of missingness) and according to the patterns of service use and EQ-5D data. In addition, we plan to do a complete case analysis as a sensitivity analysis and will compare the results to the analysis based on multiple imputed datasets, which will give an indication of how robust the cost-effectiveness estimate is. We have added more detail on this in the sensitivity analysis table.
In addition, I believe HRG's are much more precise in terms of resource use, than patient reported length of stay etc. What is the rationale of not using HRG's/patient records? The intervention and CR costs are collected by the health professionals however I can't read the details of this data collection anywhere. E.g. table 1 reads 'number of hours' of health professionals' time, however what if they only spend 15 minutes? Would that be disregarded, and where do the unit costs from these hours come from?	To address the first part of this comment, some further text has been added to the discussion section of the paper to provide some justification for this and wider information for people from other countries: Healthcare resource use will be self-reported by participants in the trial, whilst this is susceptible to recall bias it allows us to access a greater range of data (including social care). The literature notes that self-report questionnaires are a valid approach to collected healthcare resource use [36]. Additionally, there are issues with routinely collected electronic datasets in England, such as challenges with cost and delays for data approval [37]. Self-report healthcare resource use questionnaires have been used in previous UK cost-effectiveness analyses of cardiac rehabilitation [38,39]. Regarding the cost of MCT and CR some further text has been added to clarify this will be

	collected by the research team. The unit costs will be taken from the sources listed (NHS reference costs and the PSSRU). Unit costs for staff time are reported in hours and we will cost all reported time, e.g. if 1.5 hours are reported, this will be multiplied by the unit cost for an hour. We have adjusted the text to reflect this.
The sensitivity analysis is well described, and comprehensive. One of the items is: using cases of anxiety and/or depression as measure of benefit. How is this information obtained? Self-reported or patient records? In case of the former, there must be some information bias? In case of the latter, if patient records are accessible, I'd recommend using them for cost analysis, cf above.	The sensitivity analysis table describes: Anxiety and depression will be classified using the HADS score (8-10 = mild, 11-14 = moderate and 15-21 = severe) (Stern, 2014). In our analysis we will focus on the cut-off used in the inclusion criteria for the Group MCT trial (≥ 8 on either the depression or anxiety subscale of the HADS). We have clarified that this refers to cases of anxiety and depression.

VERSION 2 – REVIEW

REVIEWER	M.Elske van den Akker-van Marle Department of Biomedical Data Sciences, Leiden University Medical Center, The Netherlands
REVIEW RETURNED	20-Mar-2020

GENERAL COMMENTS	The authors have carefully addressed the reviewer comments. I have no additional comments.
--

REVIEWER	Marie Kruse University of Southern Denmark, Denmark
REVIEW RETURNED	11-Mar-2020

GENERAL COMMENTS	I have read the paper and the response, but I am not convinced that the points I raised previously are met to a sufficient degree. Firstly, re. using administrative data/patient records vs self-report, I think the authors are underestimating the challenge of their proposed method. Patients are followed-up after 4 and 12 months, which means that they should report 8 months of GP-visits, prescription drugs, etc etc. by the last follow-up. Also, if they can't remember, their resource use is estimated in an extremely uncertain way, using the last month as guideline. On the other hand, use of electronic records are rejected because the authors need to apply for it, and because of delays which shouldn't be a problem after the study has ended. I am still convinced that the method for cost calculation is suboptimal, and I suggest another sensitivity analysis, where electronic records are used instead of self-reported health care use. Secondly, I think that mentioning intention-to-treat in a cost-effectiveness analysis also implies that you include all randomized patients, regardless of compliance. So something should be done about the missing data. I am not convinced by the argument in favour of missing-at-random. The complete case analysis is a major improvement, but I would suggest also yet another sensitivity analysis altering the MAR assumption. If data are
---

	indeed MAR, use data to prove it (should be from electronic records). Finally, and related, the authors have clarified that included patients are merely eligible for CR. In my country, there is a large socio-economic difference between CR-compliant and non-compliant, and therefore those screened for anxiety and depression (and, ultimately, those participating in a trial like this) would be different in terms of socio-economics from those not screened. The trial is therefore likely to target a selected population. Where CR-eligible patients are representative of the cardiac patient population, CR-compliant are not. It probably can't be solved but I think this equity and external validity issue should be addressed.
--	--

VERSION 2 – AUTHOR RESPONSE

Reviewer 1. I have no additional comments.

Thank you to Reviewer 1 for reviewing our revised paper.

Reviewer 2.

Point 1a Firstly, re. using administrative data/patient records vs self-report, I think the authors are underestimating the challenge of their proposed method. Patients are followed-up after 4 and 12 months, which means that they should report 8 months of GP-visits, prescription drugs, etc etc. by the last follow-up.

Thank you to Reviewer 2 for taking the time to review our revised paper. We have addressed the comments in the sections below.

We agree that there are considerable challenges with using self-report service use data. We would like to highlight the processes that we put in place to strengthen the questionnaire and reduce recall bias and missing data:

- The questionnaire was developed from other service use questionnaires that were used successfully in economic evaluations led by the Lead Health Economist (Linda Davies) in a mental health and multimorbidity setting.
- The questionnaire was reviewed by the patient and public involvement group and piloted in an earlier feasibility and pilot trial to check that it was feasible to complete.
- At baseline, the questionnaire was completed by the participant with assistance from a researcher. This meant participants had experience of completing it once with help before they needed to complete it unassisted. At follow-up participants were aware they could complete the questionnaire with a researcher if needed.
- The one-month recall was intended as a simple prompt to help support participants and to assist with routine visits. In previous trials research assistants remarked that it was useful in helping to prompt participants (e.g. thinking month by month rather than for the full period in one go).

We would like to highlight a few of our most recent studies in which we have utilised the same method of data collection for service use and for which publications are available:

- Lovell K, Bee P, Bower P, Brooks H, Cahoon P, Callaghan P, Carter L-A, Cree L, **Davies L**, Drake R, Fraser C, Gibbons C, Grundy A, Hinsliff-Smith K, Meade O, Roberts C, Rogers A, Rushton K, Sanders C, **Shields G** and Walker L. Training to enhance user and carer involvement in mental health-care planning: the EQUIP research programme including a cluster RCT. Programme Grants Appl Res 2019;7(9).
- Morrison AP, Pyle M, Gumley A, Schwannauer M, Turkington D, MacLennan G, Norrie J, Hudson J, Bowe S, French P, Hutton P, Byrne R, Syrett S, Dudley R, McLeod HJ, Griffiths H, Barnes TR, **Davies L**, **Shields G**, Buck D, Tully S, Kingdon D. Cognitive-behavioural therapy for clozapine-resistant schizophrenia: the FOCUS RCT. Health Technol Assess. 2019 Feb;23(7):1-144.

- Camacho EM, **Davies LM**, Hann M, Small N, Bower P, Chew-Graham C, Bagueley C, Gask L, Dickens CM, Lovell K, Waheed W, Gibbons CJ, Coventry P. Long-term clinical and cost-effectiveness of collaborative care (versus usual care) for people with mental-physical multimorbidity: cluster-randomised trial. *Br J Psychiatry*. 2018 Aug;213(2):456-463.
- Jones S, Riste L, Barrowclough C, Bartlett P, Clements C, **Davies L**, Holland F, Kapur N, Lobban F, Long R, Morriss R, Peters S, Roberts C, Camacho E, Gregg L, Ntais D. Reducing relapse and suicide in bipolar disorder: practical clinical approaches to identifying risk, reducing harm and engaging service users in planning and delivery of care – the PARADES (Psychoeducation, Anxiety, Relapse, Advance Directive Evaluation and Suicidality) programme. Southampton (UK): NIHR Journals Library; 2018 Sep.

These studies (in mental health populations or multimorbidity) have utilised similar service use questionnaires successfully and the team has extensive experience in cleaning, costing and analysing this. Nevertheless, we agree that recall bias cannot be completely eliminated and reliance on self-report service use data is a limitation. We have included this point in the discussion (please see additions to text in point 2 below).

Point 1b Also, if they can't remember, their resource use is estimated in an extremely uncertain way, using the last month as guideline.

Thank you for pointing this out. On reflection, for the primary analysis, we will treat participants with 1-month but not the full assessment period service use data as having missing data. The missing data will be imputed using the multiple imputation process defined for all participants with missing data. We will add sensitivity analyses to explore the impact of using the 1-month recall data. The first will assume that service use is equally distributed over the assessment period and that the 1-month data can be multiplied up to 4 or 8-month follow up. The second sensitivity analysis will add the 1-month cost estimates to the multiple imputation process, depending on the pattern of missing data and sufficient participants with both 1-month and 4 or 8-month cost estimates.

To address the two comments above we have edited the methods section to read:

Resource use and costs

Direct costs will be estimated for the primary analysis. Service use included in the economic evaluation is listed in Table 1. With the exception of intervention and usual care cardiac rehabilitation, data on the healthcare resources used for each participant were collected from an economic patient questionnaire at baseline and follow-up (4 and 12 months). The questionnaire was adapted from surveys used by the authors in previous trials [21–24]. The changes to the questionnaire were developed with the service user group and the clinical experts in the research team. The questionnaire was pilot tested before use. At baseline this is completed by the participant with assistance from a researcher. This means participants have experience of completing it once with help before they need to complete it unassisted. At follow-up participants complete the questionnaire on their own but were aware a researcher could help them complete the questionnaire if needed. A copy of the questionnaire is provided in the supplementary material and has been uploaded to the Database of Instruments for Resource Use Measurement (DIRUM) [21]. The questionnaire asks patients to report service use for the follow-up period (time since the last questionnaire or three months at baseline) and for the last month. If a participant struggles to recall service use for the full follow-up but knows the last month, this is included to help prompt the participant to recall service use over the full follow up period. For the primary analysis, participants with 1-month but not the full assessment period service use data will be treated as having missing data. The missing data will be imputed using the multiple imputation process defined for all participants with missing data. Two sensitivity analyses will explore the impact of using the 1-month recall data (see Table 2).

Discussion (revised paragraph 4)

Healthcare resource use will be self-reported by participants in the trial. The literature notes that self-report questionnaires are a valid approach to collecting healthcare resource use [36,37] and were used in previous UK cost-effectiveness analyses of cardiac rehabilitation [38,39]. However, self-report forms are susceptible to recall bias and missing data. To improve feasibility and reduce bias, the service use questionnaire was adapted from those used in previous trials (refs) and changes made jointly with service users and clinical experts. The forms were piloted tested and refined. Service users complete the form at baseline by interview, to familiarise them with the forms. At follow up, participants are also offered telephone assistance with completing the forms. Nevertheless, reliance on self-reported service use to estimate costs is a limitation of the economic analysis. It is not clear whether this will lead to

systematic under or over reporting of costs and/or contribute to non-systematic inaccuracy of reported service use and costs.

Point 1cOn the other hand, use of electronic records are rejected because the authors need to apply for it, and because of delays which shouldn't be a problem after the study has ended. I am still convinced that the method for cost calculation is suboptimal, and I suggest another sensitivity analysis, where electronic records are used instead of self-reported health care use.

As noted above, we share the concerns about the use of self-reported data and also explored the feasibility of using electronic records and hospital case note data when designing the trial for the funding application. Although the reviewer understandably focuses on cost and delays, the availability of such data is also an issue. We have expanded on these below and briefly within the paper. For the reasons below, we felt that self-report questionnaires were the most feasible way of collecting the full range of service use data for each participant, within the funding and time constraints of the study. We agree that a sensitivity analysis to include electronic data would provide useful insights about the cost estimates. However, we are not in a position to access the electronic data and include this analysis.

- Our study takes an NHS and social care perspective and subsequently primary, community and social care are key aspects of service use. Data linkage between hospital data and primary, community and social care is not routinely available. Accordingly, for the UK, Franklin et al (2019) recommend using service user questionnaires to collect all but hospital data. There is a substantial cost involved with either requesting the data from NHS Digital or through researcher time accessing hospital record data. Using electronic data for hospital care plus service use questionnaires for primary, community and social care increases the time and financial cost of data collection considerably.
- There are two routes for accessing electronic hospital data in the UK. The first is via NHS Digital, which includes a central repository for hospital episode statistics. Our study is part of a 5-year programme of research. At the time of the funding application, NHS Digital was paused following data protection issues. Accordingly, it was not possible to include use of this resource in the funding application or the planned timelines for the programme. We explored the feasibility of using NHS Digital again before the start of this study. The length of the delay to obtain the data was uncertain, but in excess of 12 months after the end of the trial to ensure all records were updated. In addition, we could not start the application process for the data until close to the end of the trial, adding further time to gain the stringent ethical approval required to access central electronic data, for named patients. This was not feasible in terms of the funder deadlines for completion of the trial and submission of the final report.
- We piloted using local hospital records to access participant data about use of hospital services. In the UK, services users are able to access care from more than one hospital. Importantly, service users need to be asked which hospitals they have used at each follow up point. This means there is a substantial time and financial cost involved in identifying the relevant hospitals used by each participant at each follow up point, gaining separate governance approvals to access records for each of these hospitals and then reviewing the hospital records for each participant.

We have edited the Discussion section to summarise the issues above and hope that this helps to address the reviewers' concerns:

Collecting some or all of the service use data from routinely collected electronic datasets in England was explored. Issues encountered included the need for secondary, primary, community and social care data for the study participants. Linked electronic data across these sectors is not typically available in England at local or national level. Ethical and governance approval and funding constraints meant it was beyond the scope of this study to identify all services used by participants, and review the case notes held by each service the participants used. Centrally collected databases in England, such as NHS Digital, typically do not include linked data for the range of services included in the study. This meant that the majority of service use data would need to be collected directly from participants [37]. Access to hospital service use data for named patients via NHS Digital was paused and not available when the funding application was made. At the point at which the data collection methods needed to be finalised, the time to access data from NHS Digital was estimated at 12 months from the last participant completing the final scheduled follow up to ensure the records were complete. Accordingly,

whilst collecting service use data from participants is a limitation of the economic analysis, sourcing some or all of this data was not feasible within the funding and time constraints for this study.

Reviewer 2. Point 2

Secondly, I think that mentioning intention-to-treat in a cost-effectiveness analysis also implies that you include all randomized patients, regardless of compliance. So something should be done about the missing data. I am not convinced by the argument in favour of missing-at-random. The complete case analysis is a major improvement, but I would suggest also yet another sensitivity analysis altering the MAR assumption. If data are indeed MAR, use data to prove it (should be from electronic records).

We agree that the assumption that the data are missing at random (conditional on observed responses to clinical and economic measures and baseline covariates) may not be valid. As the reviewer notes, we cannot test the assumption of MAR from observed responses and covariates. As noted above, complete, electronic data on hospital costs is not a feasible option for this study. We propose adding an additional sensitivity analysis to explore the impact of the assumption further (Faria et al, 2014; National Research Council 2010).

We have added the following text to the methods:

Multiple imputation (MI) will be used to impute missing follow up data for costs and QALYs and assumes that the data are missing at random (conditional on observed responses to clinical and economic measures and baseline covariates); MI of both costs and QALYs is recognised as an appropriate approach to deal with missing observation and missing follow-up data [22]. Missing values will be imputed for each follow up point. All missing follow up cost and utility data will be assumed to be missing at random. To make best use of available data, we plan to impute costs by category (inpatient, outpatient and primary/community care) and utility by individual EQ-5D domain (mobility, self-care, usual activities, pain/discomfort and anxiety/depression). A balance is required between including all items and the stability of the imputation model if there is a high level of missing data. The stability of the model reduces as the number of variables with missing data increases. If feasible, all individual items in each cost category will be included in the imputation model. Total costs and QALYs (baseline to follow up) will be generated using STATA passive estimation commands. The final imputation plan will be refined following a review of missing data (exploration of whether there are any patterns of missingness) and according to the patterns of service use and EQ-5D data. Two sensitivity analyses will explore whether the results vary by approach taken to missing data. The first will use complete case analysis. The second will use a pattern mixture approach that assumes compliance and clinical outcomes are systematically worse in people with missing or with incomplete follow up [28,29]. The specification of the analysis will be finalised after review of the complete and missing data patterns and discussion with clinical experts about plausible assumptions. If appropriate this analysis will identify the threshold values of compliance and clinical outcome that would lead to a change in conclusions about the relative cost-effectiveness of the within trial economic analysis.

We have added the following text to Table 2:

The pattern mixture approach will assess the impact of assuming that compliance and clinical outcomes are systematically worse in people with missing or with incomplete follow up. It will be assumed that these differences will also lead to systematic worse health benefits and higher costs for participants with missing data [28,29].

We have added the following text to the discussion:

There are a number of anticipated challenges of the work that may result in limitations associated with the economic evaluations. Regarding the within-trial analysis missing data poses a threat to economic evaluations. However, imputation methods are planned to reduce the impact of this and a complete-case analysis will be conducted as part of the scenario analysis [35]. In addition, the final publication

will report the potential impact of missing data, for example, summarising the level of missingness by data type. Nevertheless, the multiple imputation approach assumes that the data are missing at random (conditional on observed responses to clinical and economic measures and baseline covariates), which may not be valid. A high level of complete data on the primary clinical outcome, is anticipated, which may help to strengthen analysis based on the missing at random assumption. However, it may not be sufficient to account for systematic differences in costs and health benefits between participants with and without missing data or incomplete follow up. Sensitivity analysis using (i) complete case analysis and (ii) a pattern mixture approach, which assumes compliance and clinical outcomes are systematically worse in people with missing or with incomplete follow up (Faria et al, 2014; National Research Council 2010).

Reviewer 2. Finally, and related, the authors have clarified that included patients are merely eligible for CR. In my country, there is a large socio-economic difference between CR-compliant and non-compliant, and therefore those screened for anxiety and depression (and, ultimately, those participating in a trial like this) would be different in terms of socio-economics from those not screened. The trial is therefore likely to target a selected population. Where CR-eligible patients are representative of the cardiac patient population, CR-compliant are not. It probably can't be solved but I think this equity and external validity issue should be addressed.

This is an important issue, though outside of the scope of this trial. We have added some text to the discussion to address this:

As noted in the introduction, only around half of those offered CR in the UK attend and there is insufficient data available about the proportion specific to those with symptoms of anxiety and/or depression [1]. This analysis is limited to people who attend cardiac rehabilitation. Important questions remain around how to increase attendance and what methods would be cost-effective in doing so, which are outside of the scope of this analysis but should be addressed by future studies.

VERSION 3 – REVIEW

REVIEWER	Marie Kruse University of Southern Denmark, Denmark
REVIEW RETURNED	20-May-2020
GENERAL COMMENTS	I had one major reservation regarding this study, and while service use remains self-reported, the authors have thoroughly addressed the reasons for this and the obstacles for obtaining electronic records for the purpose. Other than that weakness, I think it's a sound and relevant study and should be published.